# Enhanced Net Channel Based-Heat Sink Designs for Cooling of High Concentration Photovoltaic (HCPV) Systems in Dammam City

Fahad Ghallab Al-Amri [1,*], Taher Maatallah [1], Richu Zachariah [2], Ahmed T. Okasha [1] and Abdullah Khalid Alghamdi [1]

1   Department of Mechanical and Energy Engineering, College of Engineering, Imam Abdulrahman Bin Faisal University, Dammam P.O. Box 1982, Saudi Arabia; tsmaatallah@iau.edu.sa (T.M.); ahmed.okasha1996@gmail.com (A.T.O.); abdullahkha.gh@gmail.com (A.K.A.)
2   Department of Mechanical Engineering, Amal Jyothi College of Engineering, Kanjirappally, Kottayam 686518, Kerala, India; richuzachariah@gmail.com
*   Correspondence: fgalamri@iau.edu.sa; Tel.: +966-5-0495-5412

**Abstract:** In this study, enhanced net channel based heat sink designs for cooling HCPV systems at geometrical concentration ratios ranging from 500× to 3000× are presented. The effect of increasing the number of layers in the parallel flow net channel, as well as the fraction of the coolant mass flow rate in the counter flow net channel, on the overall performance of the HCPV systems, are investigated. The various configurations of each proposed net channel based-heat sink design are examined, and a comparative analysis between the different proposed designs is performed under the climate weather conditions of Dammam city, Saudi Arabia. On one hand, the double-layered counter flow net channel heat sink outperformed the other designs in terms of electrical efficiency and in keeping the solar cell operating well below the safe operating limits, achieving a reduction in maximum cell temperature relatively compared to the parallel flow net channel with five layers and conventional mini channel of 11.72% and 12.01%, respectively. On the other hand, for effective usability of the heat recovery rate by the cooling mechanism, the parallel flow net channel is the most appropriate design since it has recorded 27.55% higher outlet water temperature than the double-layered counter flow net channel.

**Keywords:** high concentration photovoltaic; mini channel; parallel flow net channel; counter flow net channel

## 1. Introduction

High concentration photovoltaic systems represent the future of solar energy in terms of mitigating the damage caused by conventional fossil fuels, as higher electric power can be produced in a smaller area. The concentrating photovoltaic systems can save the cost of the semiconductor materials by using advanced and innovative optical devices [1,2]. However, because the solar cell is a semiconductor device, higher concentrations of the global direct normal irradiance result in increased heat flux, lowering the device's electrical efficiency and life span [1,2]. To some extent, the temperature uniformity of the solar cell influences the electrical efficiency [3]. Thermal management of HCPV systems is thus challenging and a difficult concern. As a result, the researchers' primary goal is to cool the HCPV systems with the least amount of both materials and power (pumping power of cooling fluid).

To increase the electrical performance of high concentration photovoltaic (HCPV) systems, researchers are developing several passive and active cooling technologies. Passive cooling systems employ air, liquid immersion, heat sinks, heat pipes, phase change materials, etc. which do not use external energy supply since the density-gradient force (natural convection) is the driving force of the heat transport [2–4]. However, once the concentration

ratio exceeds $10\times$, forced cooling or active cooling solutions become compulsive. Forced circulation of air or water through tubes, ground coupled central panel, earth water heat exchanger, mini and micro channels, jet impingement, etc. are examples of active cooling systems [2–4].

The easiest approach to disperse heat from CPV systems is by passive cooling with flat plate heat sinks. Flat plate heat sinks are often used in CPV systems with concentration ratios less than $500\times$ [5,6]. Valera et al. [7] studied the feasibility of the same mechanism under concentration ratios ranging from $2000\times$–$10,000\times$. Solar cell downsizing is a critical factor to be addressed for appropriate thermal management at extremely high concentration levels. Furthermore, for better thermal management, larger heat sink substrate thickness and area are necessary, but unfortunately the extracted heat cannot be utilized for other purposes. When cost is considered, aluminum heat sinks are shown to be superior to copper heat sinks. According to Alzahrani et al. [8], a micro-finned heat sink has a 22–25% higher heat dissipation than a flat plate heat sink. Aldossary et al. [9] investigated the technical feasibility of cooling CPV systems with passive and active systems in severe environments with ambient temperatures approaching $50\,°C$. Heat sinks with round pin fins and straight fins were studied for both passive and active cooling, but passive cooling was insufficient to keep the cell's operating temperature below $80\,°C$ at a concentration ratio of $500\times$, whereas active cooling can provide better electrical performance with lower cell temperature.

Zaghloul et al. [10] studied passive cooling of a triple-junction solar cell using a straight fins heat sink. Concentration ratio was determined to have the greatest impact on system performance, followed by ambient temperature and fin length. The solar cell was able to function at a higher concentration ratio of $1250\times$ using heat sinks, providing an electrical output of $43.4\,W/cm^{-2}$ compared to the uncooled cell of $2.5\,W/cm^{-2}$ which is limited to work at up to $72.5\times$. So, it is obvious that cooling CPV systems is required to provide improved electrical performance for higher concentration ratios.

The performance of several pin fin designs (in-line cylindrical, staggered cylindrical, inline conical, and staggered conical) for cooling UHCPV systems was evaluated by AlFalah et al. [11]. It was demonstrated that at a concentration ratio of $2000\times$ and a Reynolds number of 428, inline conical pin fins had the highest overall efficiency of all designs, which was around 80.20%. However, inline cylindrical pin fins produced the highest water outlet temperature of $66.16\,°C$ with an overall efficiency of 72.5%, making it appropriate for use of the extracted heat in desalination applications.

AlFalah et al. [12] investigated the performance of three types of pin fins heat sink (normal, cylindrical, and circular) and compared them with the conventional flat-plate heat sink. With a reduction of 23.28% in the maximum operating cell temperature for cell area of $1 \times 1\,mm^2$, the circular pin-fins heat sink demonstrated the best thermal performance, allowing the solar cell temperature to remain within its safe operating range even after $10,000\times$. At $8000\times$, the flat-plate heat sink costs was 14.7% more than the normal pin-fins heat sink, which is considerably worse at higher concentrations. At different concentration ratios and Reynolds numbers, Maatallah[13] tested five pin-fin heat-sink topologies for thermal performance and hydraulic resistance. With a consistent distribution of circular axial grooves (plugs) along each pin, the inline pin-fins heat sink achieved the lowest average cell temperature for all Reynolds numbers and can ensure a safe working temperature of $80\,°C$ for the solar cell up to $15,000\times$ concentration ratio and an ambient temperature up to $35\,°C$. AlFalah et al. [14] examined the performance of rectangular fins, elliptical fins, diamond pin-fins, and traditional round pin-fins heat sinks for HCPV systems with concentration ratios ranging from $200\times$ to $1200\times$. With a maximum solar cell temperature of $81.4\,°C$ and a water temperature rise of $14.21\,°C$ under $1200\times$ and a water inlet velocity of $0.02\,m/s^{-1}$, the rectangular fins heat sink outperformed the others in terms of electrical and thermal performance, but higher pumping power was required due to a higher frictional pressure losses of around $78.74\,Pa$.

Abo-Zahhad et al. [15] analysed four different configurations of confined jet impingement heat sinks for a concentration ratio of $1000\times$. Thermal stress has been reduced signifi-

cantly when coolant mass flow rate was increased. Another study, Abo-Zahhad et al. [16], confirmed that hybrid jet impingement/microchannel hybrid cooling outperforms the conventional jet impingement cooling. Torbatinezhad et al. [17] studied jet impingement/ mini-channel heat sinks and investigated the influence of pin fins angle for various mass flow rates and concentration ratios. Maximum electrical efficiency of 29% and thermal efficiency of 80% were achieved for the CPV system with a reduction in heat sink temperature for a specific mass flow rate of $0.15\,\mathrm{kg/s^{-1}}$. Increased fin angle from 0 to 40° resulted in an electrical efficiency gain of 28.95%.

Al Siyabi et al. [18] have done an experimental study to analyze the effect of a micro-channel heat sink with multiple layers. The results showed that by increasing the number of layers from 1 to 3 the electricity generation has increased by 10% with a reduction of $3.15\,°\mathrm{C}$ at a fluid flow rate of $30\,\mathrm{mL/m^{-1}}$. Thermal energy accounted for a sizable portion of the total extracted power, accounting for around 74.9% of the total energy. Elqady et al. [19] investigated a double layered micro-channel heat sink using ethanol as a coolant for CPV systems at concentration ratios ranging from $5\times$ to $20\times$. When the coolant mass flow rate is ranged between 200 and $1200\,\mathrm{mL/h^{-1}}$, the counter flow operation produced the best temperature uniformity index compared to the parallel flow orientation. Ali et al. [20] investigated four quadrant micro-channel based heat sink designs to enhance the temperature uniformity in HCPV module. For a concentration ratio of $1000\times$, the counter flow arrangement performed better compared to the parallel flow with a temperature non-uniformity of $3.1\,°\mathrm{C}$.

According to Ahmed et al. [21], water outperforms ethylene glycol, water-ethylene glycol mix, and syltherm oil 800 in terms of temperature distribution, thermal, and electrical efficiency. Researchers have also demonstrated that nano fluids can improve the performance of these systems [22,23]. However, one disadvantage of employing nanofluids is that it increases energy consumption for pumping [24]. Long-term usage of nanofluids might result in particle settling and blockage of flow channels along with erosion of metal surfaces [25]. So long term experimental studies should be conducted to analyze the practical feasibility of using nanofluids in micro-channels. Water continues to be an ideal fluid in terms of performance, affordability, and ease of use for large-scale implementation of various types of cooling strategies.

Much research is now focused on cooling systems for UHCPV systems with concentration ratios greater than $10,000\times$, (Ultra-high concentration level), but the experimentally attained concentration ratio for CPV systems is still less than $5800\times$ [13], so it is worthwhile to focus on better heat sink designs for CPV systems with concentration ratios ranging from $500\times$ to $3000\times$.

Straight fin heat sinks (channels) outperform round pin fin heat sinks due to their unique design, since they can maintain a lower operating temperature for the PV surface through greater heat dispersion [9]. So in this work mini-channel heat sinks are focused. According to Abo-Zahhad et al. [26], the channel geometry and water mass flow rate have a substantial influence on the performance of a HCPV system.

Operating at high electrical efficiency and concentration levels entails maintaining solar cell temperature well below the safe operating limit at a uniform surface temperature distribution, which is a significant challenge in the industrial-scale use of HCPV systems. The present paper is mainly focused on the active cooling mechanism at high concentration ratios ranged between $500\times$ and $3000\times$ based on different enhanced designs of net channel based-heat sink for heat extraction, that enables the breakdown of the records/limits of the conventional mini channel based-heat sink and paves the avenue for potential applications of the downstream hot water leaving the cooling system of the HCPV. Moreover, in the last section from this paper, a performance chart of the parallel-flow net channel with five layers, and the counter flow net channel with two layers is analyzed considering the climate weather conditions of Dammam city (26°23′ N, 49°59′ E).

The following is an outline of the paper. Section 2 presents the detailed geometrical design used for the analysis, explanation of the proposed design and various configurations analyzed and the methodology used in this study. Section 3 presents and discusses the

findings, which include the influence of the number of layers in the net channel, the effect of the flow rate fraction in the double-layered net channel heat sink, and a comparison of parallel and counter flow topologies. In addition, the performance of the optimum parallel and counter flow net channel based-heat sink configurations are compared and discussed under the climate weather conditions of Dammam city. Section 4 summarizes the results of this paper.

## 2. Design and Methodology

### 2.1. Geometric Designs

In the present work, three net channel based-heat sink designs are being investigated in order to improve the overall system performance and foreshadow the avenues for combining the single-cell HCPV modules with potential applications such as air conditioning, water heating, and desalination systems. The conventional mini-channel based-heat sink is shown in Figure 1, which is used as the baseline design for the sake of comparison. It is also very important to consider the scalability of an optimal design, however, this study does not investigate the optimum size dimensions of the fins (pitch wise spacing, aspect ratio, width, and length) since already optimal configuration for one set of parameters may not be optimal for another set. Furthermore, if the pumping power is not limited, as assumed in the current work, there is no optimum channel dimensions, since the larger the size is, the better the convective heat transfer would be because that will compensate the increase in bulk thermal resistance especially at small flow rates. To investigate the effect of increasing the number of layers of coolant flow on the HCPV system performance, a net channel heat sink design, as shown in Figure 2, is proposed, which is based on increasing the cooling system's heat transfer area, in other terms decreasing the overall thermal resistance. In the first discussion section, the effect of the number of layers in the net channel performance has been investigated. Four different parallel-flow net channel heat sink configurations, based on the number of water channel layers (5, 4, 3, and 2 layers), are considered as shown in Figure 2. Then, to investigate further the influence of the mass flow fraction across the net channel layers, a double-layered net channel heat sink design operating with a counter-current flow (water flowing in opposing directions in the upper and lower channels) is considered and shown in Figure 3. The coolant flow rate in the upper and lower layers of the counter flow net channel is varied in proper percentages to select the best configurations in terms of cell surface temperature and temperature uniformity.

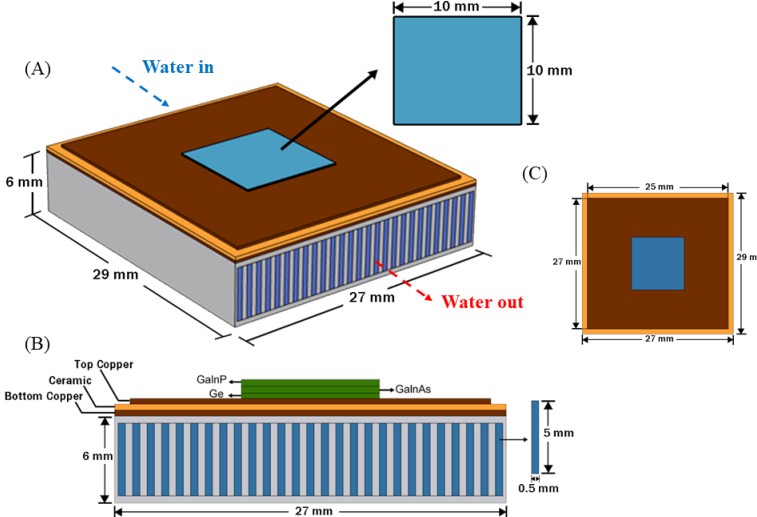

**Figure 1.** Geometric model of the conventional mini channel heat sink design (**A**) Perspective view, (**B**) Side view and (**C**) Top view.

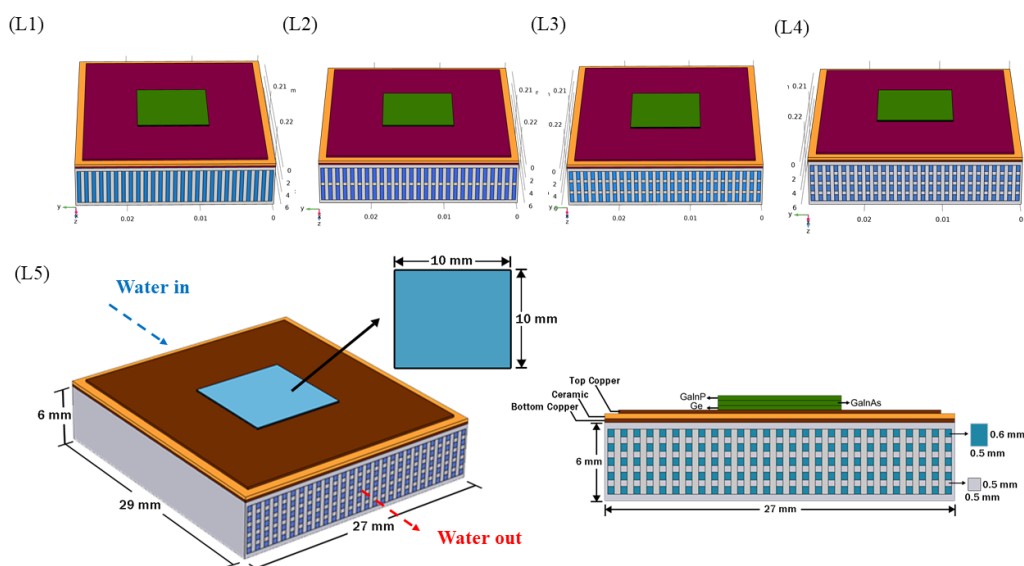

**Figure 2.** Geometric model of the net channel heat sink of aluminum heat sink (**L1**) Conventional mini channel heat sink (**L2**) Net channel heat sink with 2 layers, (**L3**) Net channel heat sink with 3 layers, (**L4**) Net channel heat sink with 4 layers, and (**L5**) Net channel heat sink with 5 layers.

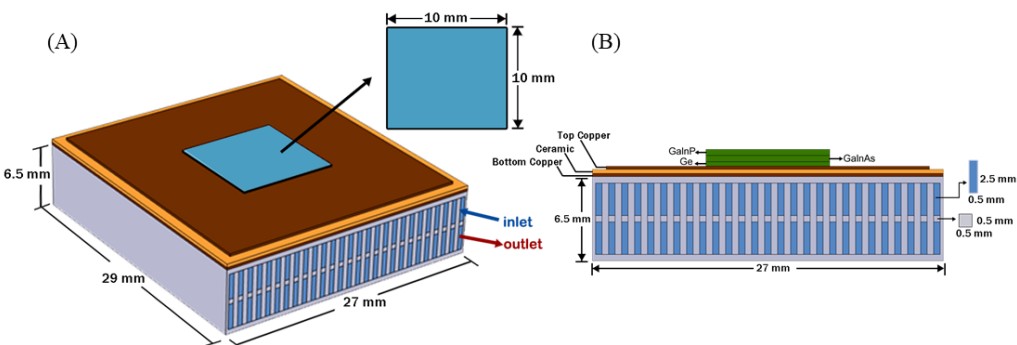

**Figure 3.** Geometric Model of the counter flow net channel heat sink with two layers (**A**) Perspective view and (**B**) Side view. The percentage of flow in the upper and lower layers are varied for different configurations of the design

The performance and practicality of all suggested net channel based-heat sink designs for a conventional triple-junctions solar cell are evaluated. The heat load is applied on the germanium layer of the multi-junction cell. This study determines the best and worst-case scenarios, as well as the temperature distribution under various mass flow rates. The overall heat sinks length, width, and height are 29, 27, and 6 mm, respectively. The first proposed design has 26 heat sink fins and a total of 27 water channels, whereas the second design with five layers has 135 water channels. Detailed dimensions of the investigated heat sink designs were presented in Figures 1–3. The effect of water mass flow rate and different cooling designs under different concentration ratios (500×–2000×) on cell temperature, thermal, and electrical efficiencies are studied at a water inlet temperature of 25 °C. All the configurations are tested for a cooling water mass flow rate of 0.001 kg/s$^{-1}$ to reduce the pressure drop across the mini channels, and maximize the residence-time of water particles inside the channels. All heat sink designs are tested under the same conditions to determine which system will be more effective in reducing the most uniform solar cell surface temperature, and yield the highest outlet water temperature at higher electrical efficiency under a concentration ratio of up to 3000× for sufficient temperature uniformity achievement.

## 2.2. Simulation Setup

In this work, the computational models of the software COMSOL Multiphysics® Version 5.6, which is based on Finite Element Analysis (FEA), are utilised to quantitatively simulate the thermal performance for the cooling of the HCPV system. The simulations were conducted on a CPU with a 8.00 GHz and 16.0 GB of RAM capacity. To validate the correctness of the obtained numerical solutions, a mesh-sensitive analysis was elaborated.

## 2.3. Governing Equations and Boundary Conditions

### 2.3.1. Governing Equations

In this study, the energy transfer through the HCPV system was introduced in Figure 4. Triple junction III–V solar cells are used in HCPV technology due to their higher efficiency, wider energy band gap, better responsiveness to higher concentration levels, and lower degradation temperature coefficients [9]. So, the HCPV module consists of triple-junction solar cell (1) Germanium (Ge), (2) Indium Gallium Phosphide (GaInP), (3) Indium Gallium Arsenide (GaInAs), Copper layers, Ceramic layer, and the MCBHS.

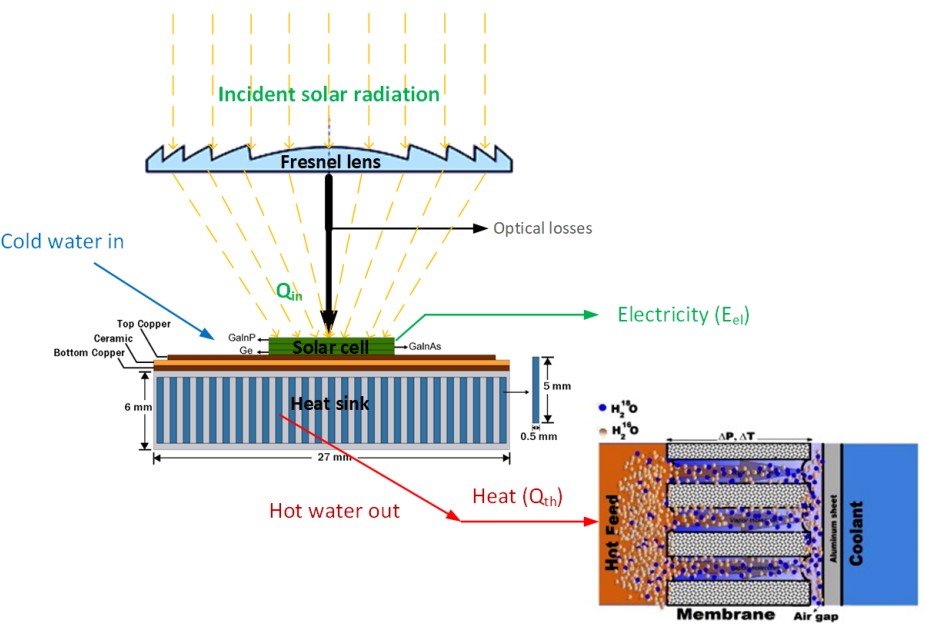

**Figure 4.** Energy transfer through the UHCPV system.

The continuity equation is written as follows:

$$\nabla \cdot (\rho \vec{v}) = 0 \tag{1}$$

The momentum conservation equation is:

$$\nabla \cdot (\rho \vec{v} \vec{v}) = -\nabla p + \nabla \cdot (\overline{\overline{\tau}}) + \rho \vec{g} \tag{2}$$

where $\rho \vec{g}$ is the gravitational body force. The viscous stress tensor can be described by:

$$\overline{\overline{\tau}} = \mu \left( \nabla \vec{v} + (\nabla \vec{v})^T \right) - \frac{2}{3} \mu (\nabla \cdot \vec{v}) I \tag{3}$$

The heat transfer equation can be written as:

$$\nabla \cdot (\rho \vec{v} E) = -\nabla (k \cdot \nabla T) + \nabla \cdot (\vec{v} \cdot \overline{\overline{\tau}}) \tag{4}$$

### 2.3.2. Overall Single-Cell HCPV Module Efficiency

The total sun power projected on the solar cell is:

$$Q_{\text{in}} = DNI \cdot CR \cdot \eta_{\text{opt}} \cdot A_{\text{cell}} \tag{5}$$

where $A_{\text{cell}}$ is the active cell surface area. Taking into account the electrical conversion efficiency of the solar cell, $\eta_{\text{sc}}$, the power yield of the solar cell is the following:

$$P_{\text{sc,elec}} = Q_{\text{in}} \cdot \eta_{\text{sc}} \tag{6}$$

while the rest power will be dissipated as unwanted rate of heat cross the solar cell boundaries as:

$$Q = Q_{\text{in}} \cdot (1 - \eta_{\text{sc}}) \tag{7}$$

The heat power carried by the fluid is:

$$Q_{\text{th}} = \dot{m} \cdot C_p \cdot (T_{\text{f,out}} - T_{\text{f,in}}) \tag{8}$$

where $\dot{m}$ is the mass flow rate (kg/s$^{-1}$) and $C_p$ is the specific heat capacity (J/kg$^{-1}$/K$^{-1}$) of the fluid. The thermal efficiency, $\eta_{\text{th}}$, can be expressed as:

$$\eta_{\text{th}} = \frac{Q_{\text{th}}}{Q_{\text{in}}} \times 100 \tag{9}$$

The overall HCPV system efficiency can be described as follows:

$$\eta_{\text{overall}} = \frac{Q_{\text{th}} + P_{\text{elec,net}}}{Q_{\text{in}}} \times 100 \tag{10}$$

The required pumping can be determined by determining the pressure drop inside the channel and for a constant density flow it can be calculated as:

$$P_{\text{pump}} = \dot{V} \cdot \Delta P \tag{11}$$

where $\dot{V}$ is the volume flow rate of the fluid (m$^3$/s$^{-1}$).

### 2.3.3. Boundary Conditions

To solve the above-mentioned equations (continuity, momentum, and energy equations), the following boundary conditions and assumptions are applied (Figure 5):

- No slip condition is applied to all the walls
- The water flow is assumed laminar, steady, and incompressible.
- The outlet pressure is equal to $P_s = P_{\text{atm}}$.
- Reynolds number is ranged between 7–508.
- The range of inlet water mass flow rate in each channel is set to be 0.001–0.006 kg/s$^{-1}$.
- The natural convection heat transfer coefficient is 15 W/m$^{-2}$/K$^{-1}$, which is reasonable for free air stream conditions.
- The sided walls of the mini channels and the bottom sheet of the heat sink are insulated.
- The ambient and inlet water temperatures are assumed equal to 25 °C.
- The thermophysical properties of the water (i.e., density, specific heat capacity, thermal conductivity...) are considered temperature-dependent scalars.

The dimensions and thermal properties of the solar cell and heat sink are recapitulated in Tables 1 and 2.

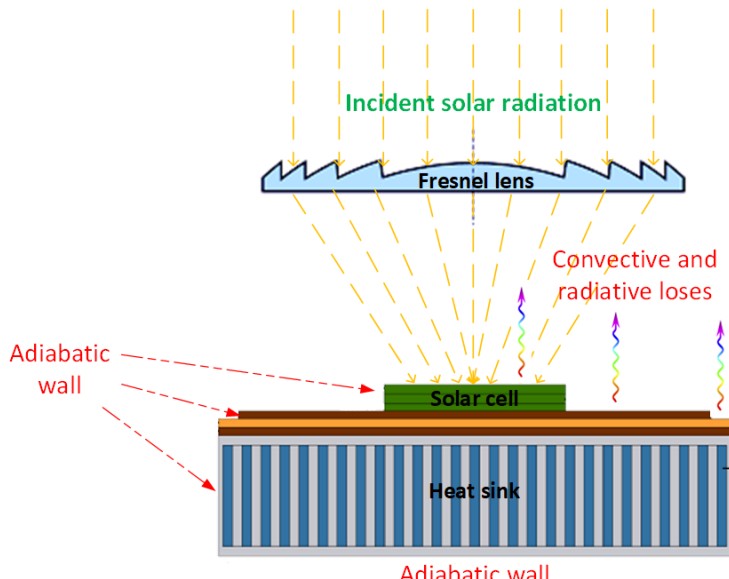

**Figure 5.** Schematic diagram of the boundary conditions of both the HCPV system and the attached heat sink.

**Table 1.** Dimensions of the HCPV layers.

| Material | Length (mm) | Width (mm) | Thickness (mm) |
|---|---|---|---|
| Aluminum | 29 | 27 | 6 |
| GaInP | 10 | 10 | 0.07 |
| GaInAs | 10 | 10 | 0.07 |
| Ge | 10 | 10 | 0.07 |
| Copper (1) | 27 | 25 | 0.25 |
| Ceramic | 29 | 27 | 0.32 |
| Copper (2) | 29 | 27 | 0.25 |

**Table 2.** Thermo-physical properties of the used materials.

| Material | k (W/m$^{-1}$/K$^{-1}$) | C (J/kg$^{-1}$/K$^{-1}$) | $\rho$ (kg/m$^{-3}$) | Emissivity ($\epsilon$) |
|---|---|---|---|---|
| Aluminum | 160 | 900 | 2700 | - |
| GaInP | 73 | 370 | 4470 | 0.9 |
| GaInAs | 65 | 550 | 5316 | - |
| Ge | 60 | 320 | 5323 | - |
| Copper | 400 | 385 | 8700 | 0.05 |
| Ceramic | 27 | 900 | 3900 | 0.75 |

### 2.3.4. Grid Independence Study

The generated mesh is a physics-controlled mesh. The mesh has been created including the mesh element types adopted to the physics that have been included in the developed model settings. This also goes for any boundary condition, domain condition or constraints applied to the model geometry. In the current study, the generated mesh has a customized number of Tetrahedral shaped-elements. Grid independence study was carried out to ensure the accuracy of the modeling cases. The test independence test has been performed for each studied case, and for the sake of illustration, the case of the conventional mini channel heat sink design is demonstrated in this section. In Figure 6, the variation of the maximum solar cell temperature is plotted against the number of mesh element sizes, which is varied from $0.177 \times 10^6$ to $14 \times 10^6$. The figure revealed that the maximum surface cell temperature has an almost constant value for the grid elements from 3.2 to 14 million elements. In addition to that, the most appropriate number of elements that represent the most accurate data and computational time was considered as 3.2 million.

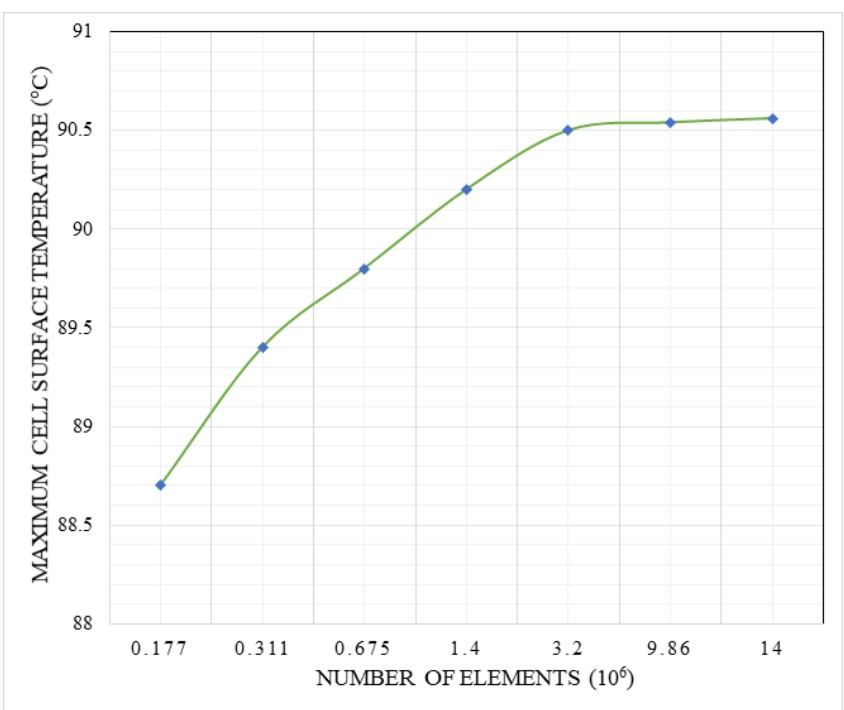

**Figure 6.** Grid independence study for the maximum solar cell temperature at a concentration ratio of 2000× and flow rate of 0.001 kg/s$^{-1}$.

### 2.3.5. Validation Study

The validation of the HCPV system was carried out and conducted against Ahmed et al. [21] to ensure the accuracy of the numerical simulation. Figure 7 depicts the tendency of the maximum solar cell temperature against different water flow rates at the same CPV module dimensions, thermophysical properties, and applied boundary conditions. Increasing the water flow rate for both Ahmed et al. [21] model and the present work has a positive effect on decreasing the maximum solar cell temperature. The obtained results based on the developed numerical model compared to the work of Ahmed et al. [21] has yielded a maximum error of about 1.6%, while the minimum error was 0.3%, confirming the validation of the validity of the numerical simulations.

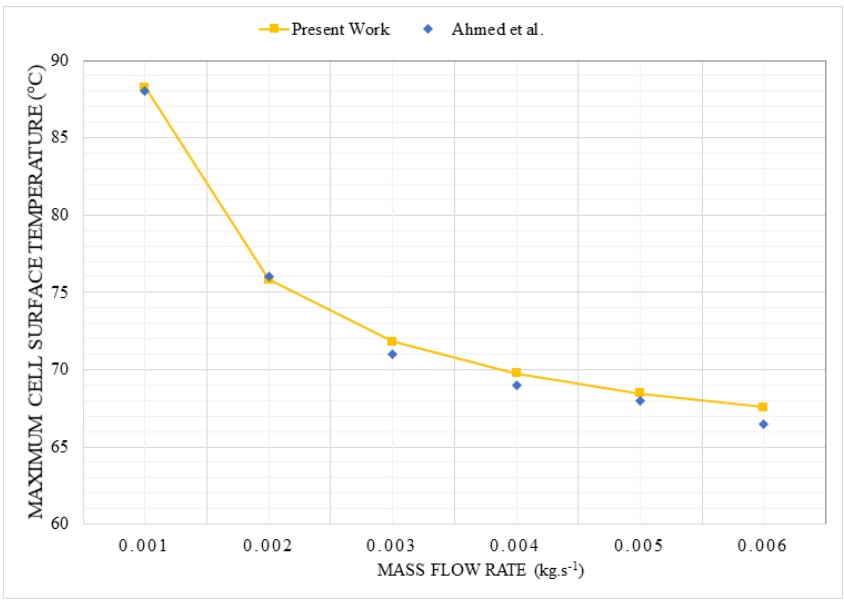

**Figure 7.** Variation of the solar cell maximum temperature against different water flow rates for the work of Ahmed et al. [21] and the present work.

## 3. Results and Discussion

The performance and the relative improvements made by each net channel-based heat sink compared to the existing conventional one is analyzed in terms of maximum solar cell temperature, surface temperature distribution across the cell, outlet water temperature, thermal resistance, and thermal and electrical efficiency of the HCPV module as function of different high concentration ratios.

### 3.1. Parallel Flow Net Channel Based-Heat Sink Configurations

The effect of increasing the number of layers on the net channels performance is studied in this section. Four different parallel flow net channels (all the inlet water streams in the different layers are entering at one end, and leave at the opposite end) are examined and compared to the conventional mini channel.

### 3.1.1. Maximum Cell Surface Temperature and Temperature Distribution

The performance of the HCPV system using water as a cooling fluid at various concentration ratios is demonstrated for the four distinct configurations of the net channel design and the conventional mini channel. The fluctuation of the maximum solar cell temperature at a mass flow rate of $0.001\,\mathrm{kg/s^{-1}}$ and with varied concentration ratios ranging from $500\times$ to $3000\times$ is shown in Figure 8. The objective of these net channel configurations is to keep the maximum measured temperature at different concentration ratios below the manufacturer's maximum recommended temperature ($110\,°\mathrm{C}$). From the recorded data the maximum temperature for all configurations was achieved by the conventional channel where the temperature of the cell reached up to $119.47\,°\mathrm{C}$ and $88.27\,°\mathrm{C}$ at concentration ratio of $3000\times$ and $2000\times$, respectively. While the best scenario and lowest temperature were observed by using the parallel flow net channel with five layers. In effect, under the mass flow rate, the reduction in maximum solar cell temperature has varied from $4.91\,°\mathrm{C}$ to $3.46\,°\mathrm{C}$ under a concentration ratio of $3000\times$ and $2000\times$, respectively. Under the same test conditions, the other configurations were ranged between the conventional and the parallel flow net channel with five layers. It is worthy to note that the mass flow rate of $0.001\,\mathrm{kg/s^{-1}}$ has been set in all numerical simulations as it has yielded the maximum water outlet temperature using mini-channel based-heat sink, which fall within the interest of maximizing the thermal energy recovered from the HCPV system that would be effective for potential applications such as desalination.

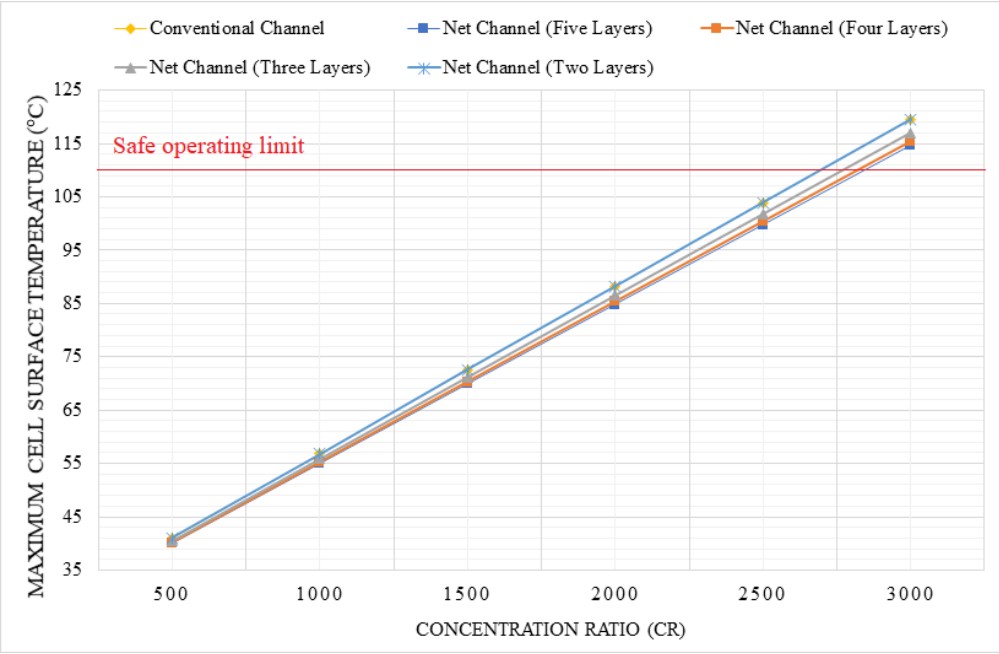

**Figure 8.** Variation of the maximum temperature of the CPV solar cell with different concentration ratio.

In addition to the maximum cell surface temperature, the temperature distribution of the solar cell, in other words the temperature uniformity, at different concentration ratios and a mass flow rate of $0.001 \, \text{kg/s}^{-1}$ is presented in Figure 9. The overall system performance of the HCPV system will improve significantly as the temperature range (difference between the maximum and minimum cell temperature) decreases. The parallel flow net channel design with five layers has shown the most uniform temperature distribution, which significantly reduced the extend of the hot spots over the cell surface area, followed by the four, three, and two-layers mini channel designs, whereas the conventional mini channel design has exhibited the worst scenario with a temperature range increase by 4.81 °C relatively compared to net channel design of five layers. The temperature range of the net channel design with five layers was ranged from 28.81 °C to 43.22 °C, under concentration ratios varying from 2000× to 3000×, respectively. So it is evident that, the new parallel flow net channel configurations outperform the conventional channel in terms of maximum solar surface temperature and temperature uniformity across the solar cell.

Figure 10 shows the contours diagram for the temperature T (in Celsius) over the solar cell and the top copper layer of the substrate for all investigated configurations at a geometrical concentration ratio of 2000× and water flow rate of $0.001 \, \text{kg/s}^{-1}$. Figure 11 shows the contours diagram for the temperature, T (in Celsius) over the solar cell and the top copper layer of the substrate for the five-layers net channel configuration at all considered geometrical concentration ratios ranging from 500×–3000× and a water flow rate of $0.001 \, \text{kg/s}^{-1}$. As the number of layers increases, all thermal resistances increase, and the flow rate at each layer decreases. This results in an increase in the bulk flow resistance. Moreover, as the number of layers increases, the fin efficiency, and the heat transfer coefficient decrease. These effects cause an increase in the convection resistance in each individual layer. Whereas, this increase in thermal resistance can be compensated by the increase in heat transfer areas (for a number of layers above two) because the thermal resistances are in parallel in the resistance network.

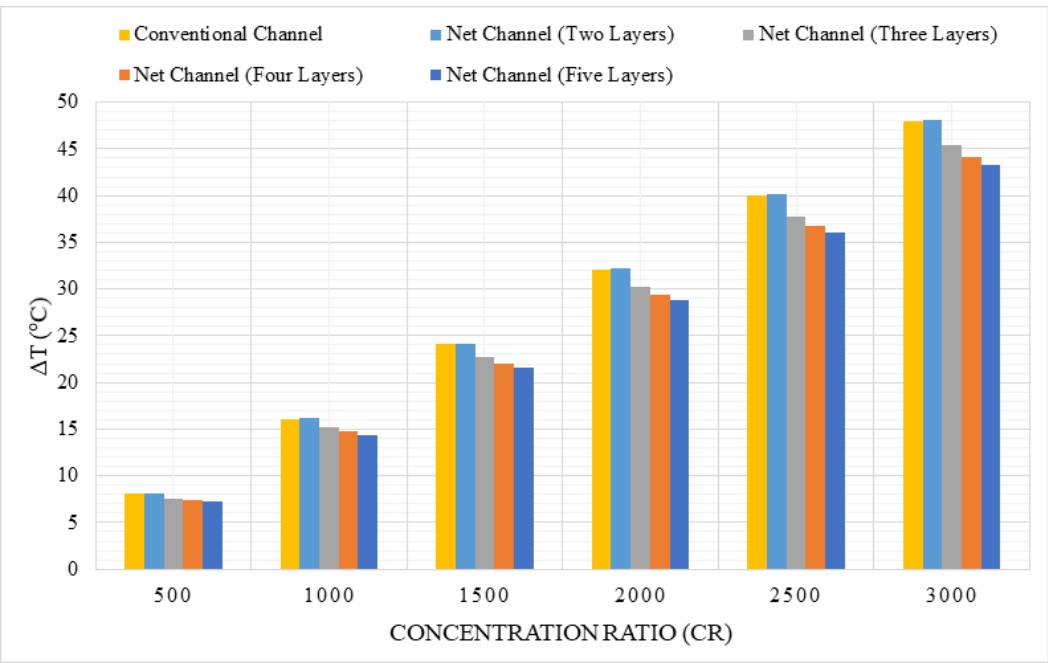

**Figure 9.** Variation of the temperature distribution of the CPV solar cell with different concentration ratio.

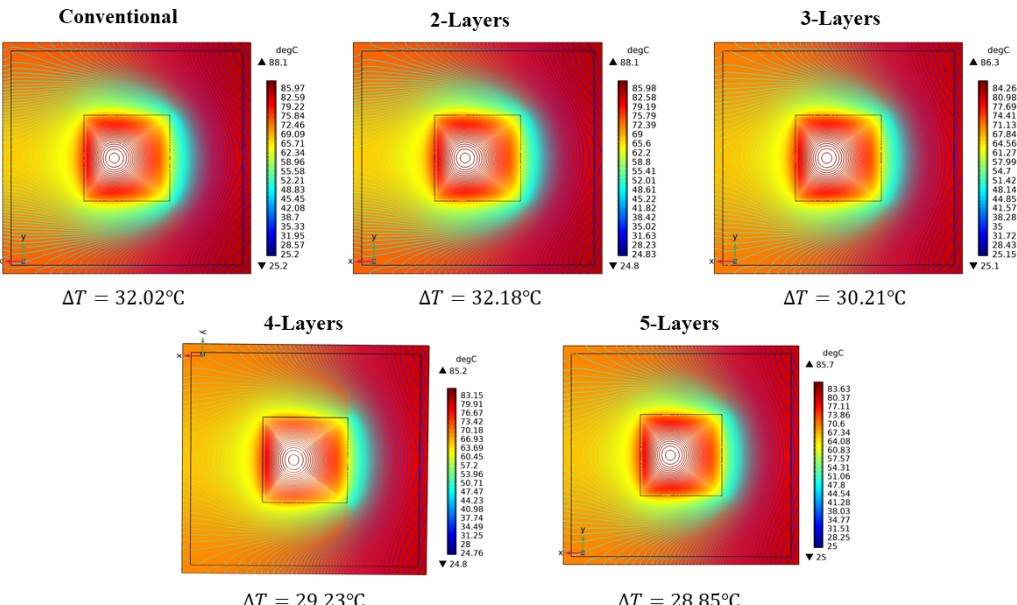

**Figure 10.** Temperature contours across the solar cell for each configuration at concentration ratio of $2000\times$ and water flow rate of $0.001\,\mathrm{kg/s^{-1}}$.

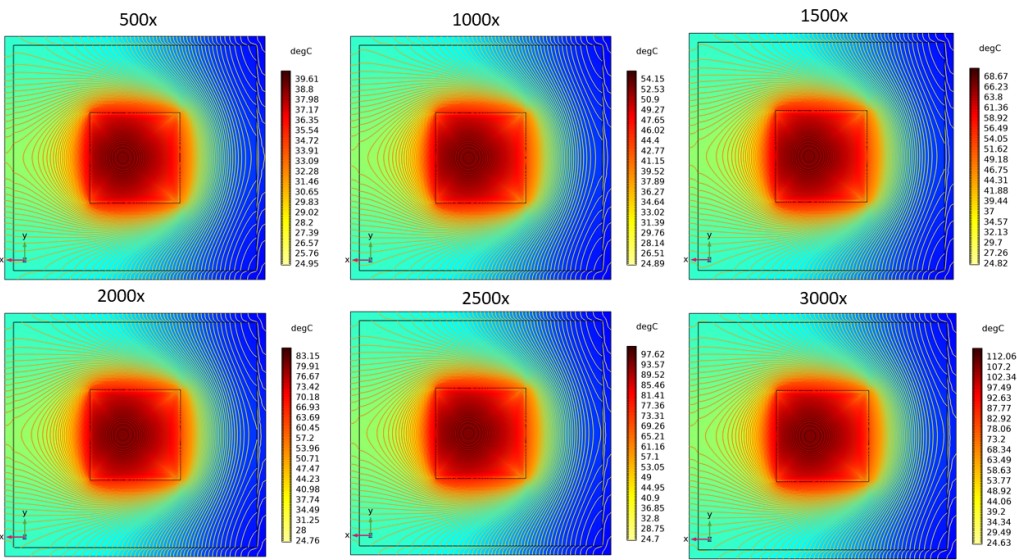

**Figure 11.** Temperature contours across the solar cell for the five-layers net channel configuration at all considered geometrical concentration ratios ranging from $500\times$–$3000\times$ and a water flow rate of $0.001\,\mathrm{kg/s^{-1}}$.

### 3.1.2. Outlet Water Temperature

The increase of the outlet water temperature results in increasing the maximum extracted energy from the solar cell and allowing more adequate range of hot feed-in water to the desalination units, then improved diffusion effect and gained output ratio. The variation of the outlet water temperature at a constant mass flow rate of $0.001\,\mathrm{kg/s^{-1}}$ with varied concentration ratios ranging from $500\times$ to $3000\times$ is shown in Figure 12. According to the observed results, the outlet water temperature was recorded to be almost constant for all the parallel flow net channel designs. The outlet water temperature for heat sink design with five layers reached up to $52.67\,°\mathrm{C}$ and $66.49\,°\mathrm{C}$, respectively, under concentration ratios of $2000\times$ and $3000\times$. This range of outlet temperature can operate effectively an air-gap membrane desalination since it falls within the range of the optimal hot feed-in water temperatures [27,28]. Furthermore, for all tested conditions, the outlet water temperature using

the conventional and all new parallel flow net channel configurations is approximately the same and close to each other. Besides that, increasing the concentration ratio had a significant effect on increasing the outlet water temperature for all configurations.

### 3.1.3. Thermal Resistance

Thermal resistance is regarded as an important factor in improving the overall performance of the HCPV system by enhancing the heat transfer between the heat sink channel and the working fluid. As thermal resistance decreases, system performance improves, resulting in effective temperature distribution for the solar cell surface, which is considered as a considerable solution to the overheating issue, and the best of heat sink performance. The variation in thermal resistance versus concentration ratios at a constant flow rate of $0.001 \, \text{kg/s}^{-1}$ is shown in Figure 13. Since the mini channel heat sink is thin and the materials used, such as silicon, have very high thermal conductivity, the conduction resistance is normally a small part of the overall resistance. For small flow rates (that has been fixed constant in each configuration), pressure drop is still small and the bulk resistance is still significant compared to the convective thermal resistance than can increase only for high flow rates even with more number of layers and higher fin efficiencies once the adiabatic fin-end boundary condition is applied. These reasons can explain the closed values of the thermal resistances, as well as the outlet temperatures at a small mass flow rates, therefore no configuration has the upper hand over the others in terms of higher outlet temperature. According to the results, the parallel flow net channel design with five layers has the lowest thermal resistance of $0.392 \, °\text{C/W}^{-1}$, while the conventional and parallel flow net channel with two layers designs have the highest thermal resistance of $0.433 \, °\text{C/W}^{-1}$ at the same flow rate.

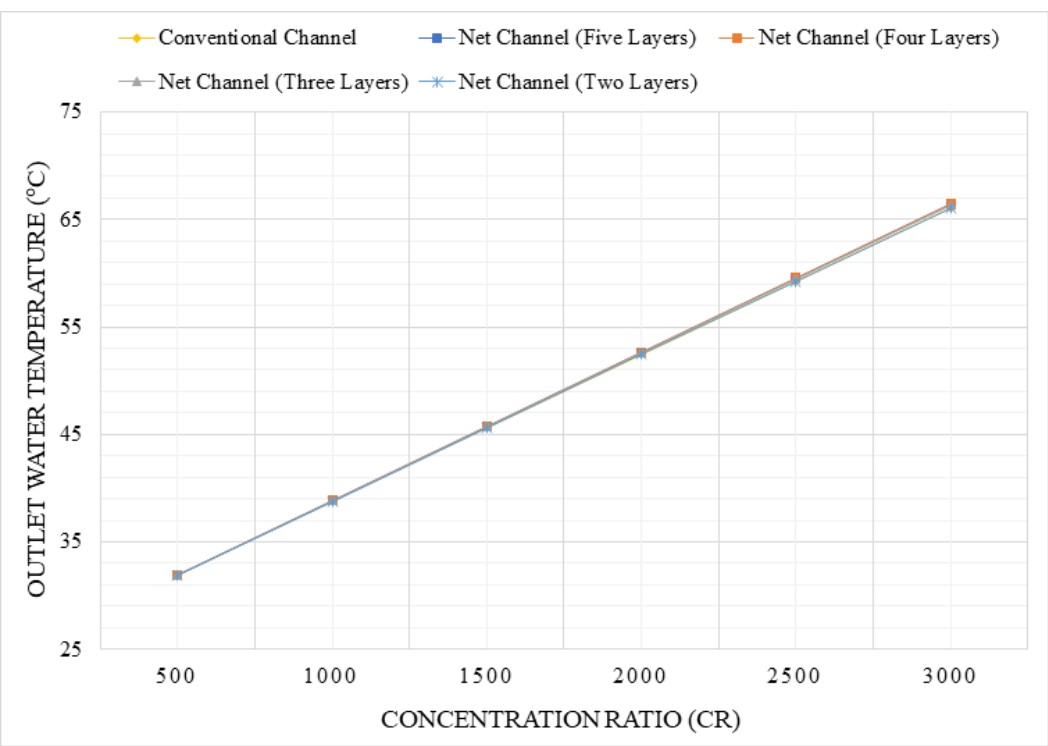

**Figure 12.** Variation of the outlet water temperature of the CPV solar cell with different concentration ratio.

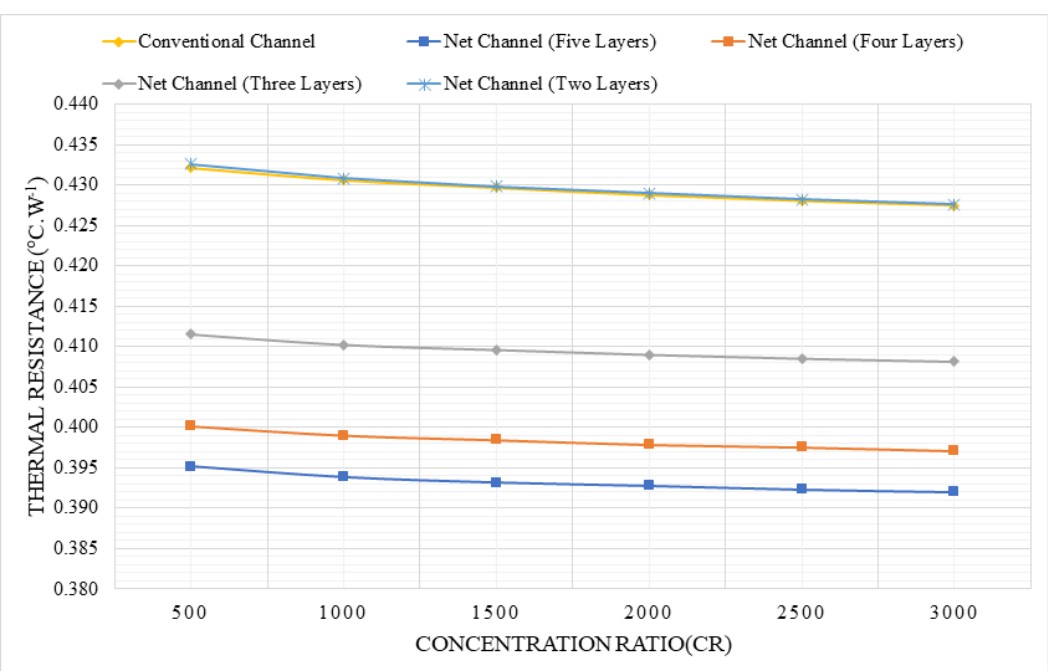

**Figure 13.** Variation of the thermal resistance with different concentration ratio.

### 3.2. Counter-Flow Net Channel Based Heat Sink Configurations

The effect of varying the flow rate fraction (%) in the upper and lower layers of the counter-flow net channel (the inlet water streams in the two layers are entering at one end, and leave at the same end) is investigated, and the results are compared to those of the conventional mini channel. In this section, only the double-layered net channel (with two layers) has been considered to shoot the light on the influence of operating the net channel layers with variable mass flow rate fractions, comparatively relative to the existing mini channel based-heat sink design operating with the same overall mass flow rate. It is worthy to notice here that such a configuration (counter-flow net channel) including more than two layers may result in far more complicated net channels, and for that reason the study has been limited to the investigation of just the counter-flow double-layered net channels. The average values of the different meteorological data for the city of Dammam is listed in the following table.

#### 3.2.1. Maximum Cell Surface Temperature and Temperature Distribution

The variation of the maximum solar cell temperature at a constant mass flow rate of $0.001 \, \text{kg/s}^{-1}$ with different concentration ratios ranging from $500\times$ to $3000\times$ is shown in Figure 14. The maximum cell surface temperature for the conventional channel rises to $119.47 \, °C$ at a concentration ratio of $3000\times$, which is higher than the safe operating limit of the solar cell, ($110 \, °C$). The counter flow net channel had the lowest cell surface temperature with equal flow rates in both layers, and the reduction in maximum cell surface temperature was significant, ranging from $15.43 \, °C$ to $22.81 \, °C$ when compared to the conventional channel at concentration ratios of $2000\times$ and $3000\times$, respectively. Furthermore, the surface temperature was kept below the recommended limit, with maximum temperatures reaching $72.83 \, °C$ and $96.66 \, °C$ at concentration ratios of $2000\times$ and $3000\times$, respectively. Under the same test conditions, the surface maximum temperature was almost the same for the other shared mass flow fractions between the two layers, with the exception of the case 90–10% flow rate.

Figure 15 depicts the temperature distribution of the solar cell for various concentration ratios at a constant mass flow rate of $0.001 \, \text{kg/s}^{-1}$. The overall system performance of the HCPV system will increase significantly when the surface temperature range of the solar cell decreases. As illustrated in the Figure 15, all five configurations of the counter flow net channel based-heat sink have an uniform temperature distribution that considerably

reduces the hot spots. The conventional mini channel design, on the other hand, performs poorly in terms of temperature distribution, with a maximum temperature of range of 7.37 °C and 10.91 °C when compared to the counter flow net channel configurations at concentration ratios of 2000× and 3000×, respectively. The lowest cell surface temperature difference was reported by the counter flow net channel configuration with 30% flow in the upper and 70% flow in the lower layer and ranged from 30.92 °C to 37.11 °C, respectively, for concentration ratios ranging from 2000× to 3000×.

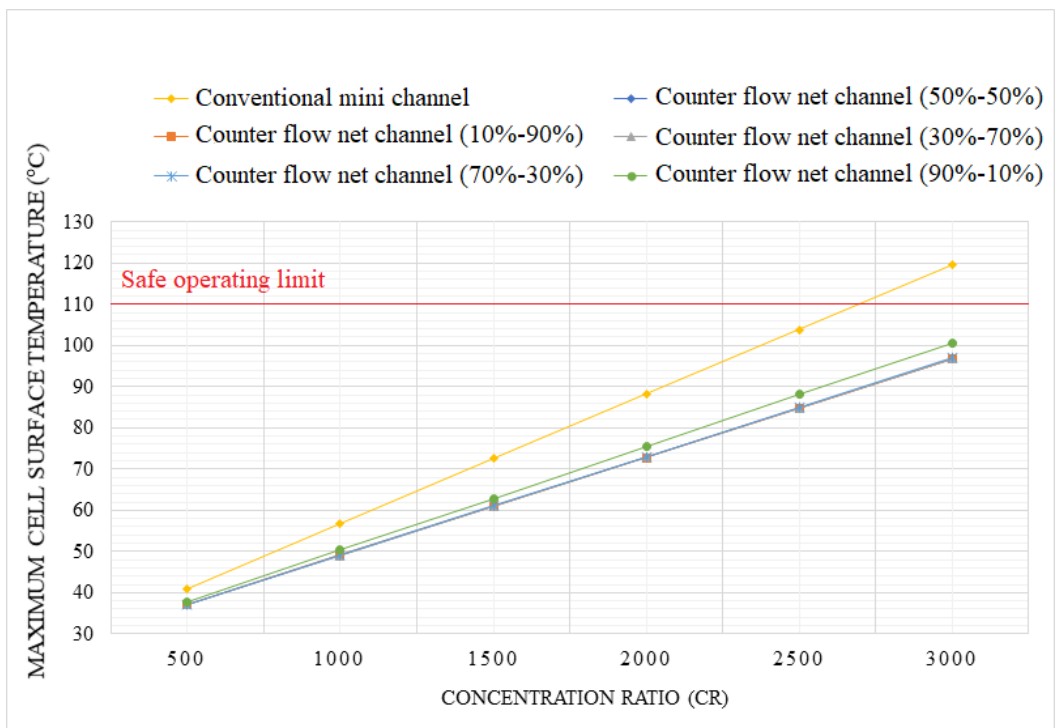

**Figure 14.** Variation of the maximum temperature of the CPV solar cell with different concentration ratio.

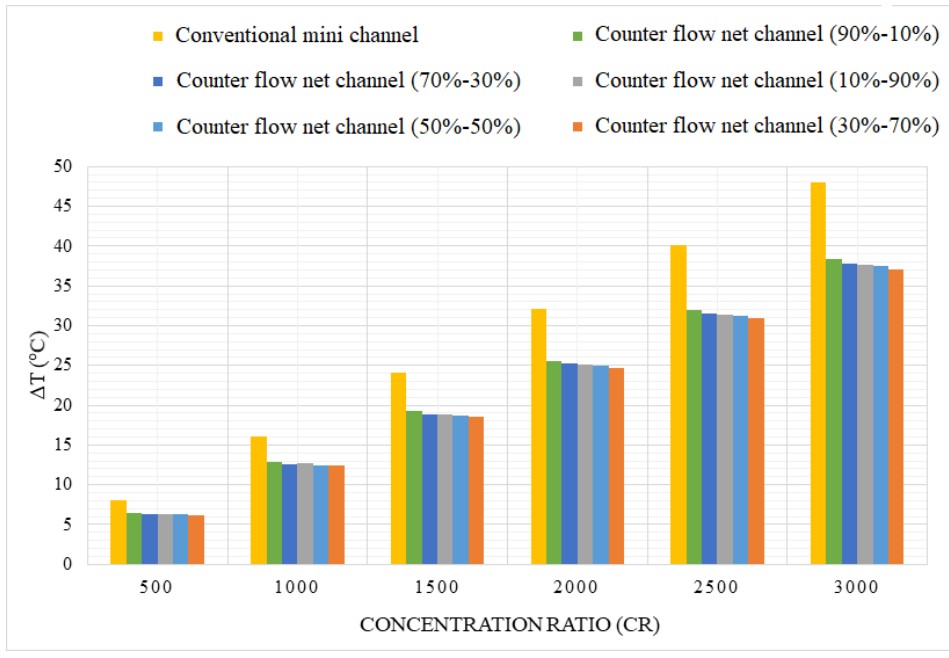

**Figure 15.** Variation of the temperature distribution of the CPV solar cell with different concentration ratio.

### 3.2.2. Outlet Water Temperature

In the case of a counter flow net channel based-heat sink design, the average outlet water temperature is very low when compared to the conventional mini channel, because the water in the upper layer of the heat sink reaches its maximum temperature in the mid-path just below the triple-junction solar cell, where the hot water has been in thermal contact with the entering cold water in the second layer at the last part of the first layer. The variation of the outlet water temperature with a constant mass flow rate of $0.001 \, \mathrm{kg/s^{-1}}$ and various concentration ratios ranging between $500 \times$ and $3000 \times$ for the counter flow net channel configurations is shown in Figure 16. The highest outlet water temperature was recorded by a counter flow net channel configuration with a flow rate of 90% and 10% in the upper and lower layers, respectively, as shown in the Figure 16. The outlet water temperature for this configuration reached up to 33.27 °C and 37.38 °C for concentration ratios of $2000 \times$ and $3000 \times$, respectively. Furthermore, raising the concentration ratio showed a significant influence on increasing the exit water temperature for all configurations.

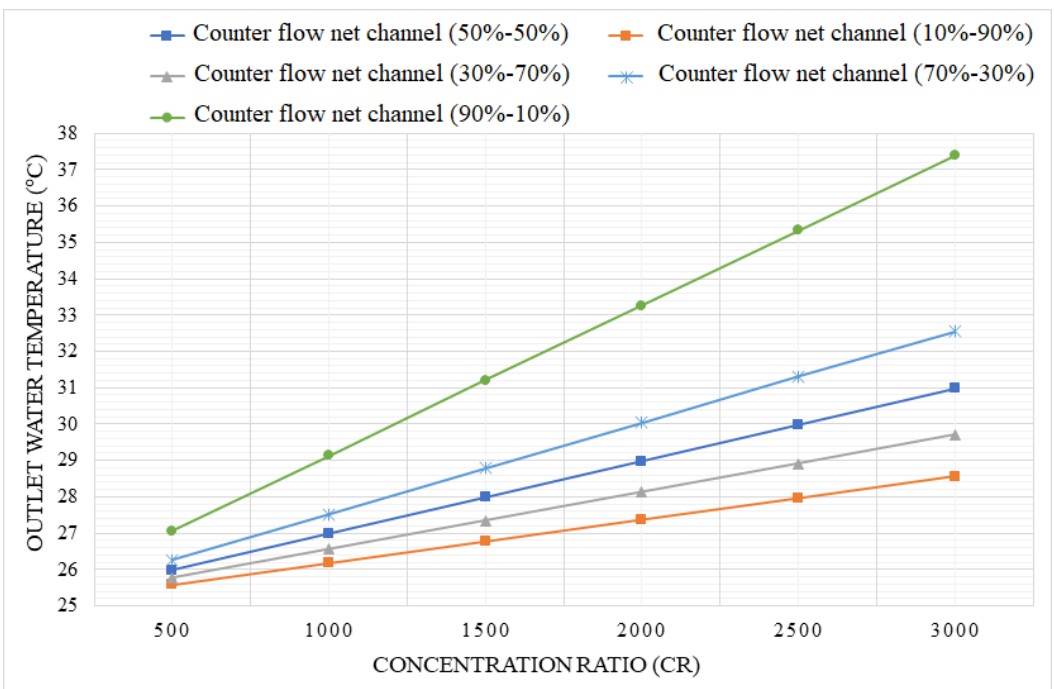

**Figure 16.** Variation of the water outlet temperature of the CPV solar cell with different concentration ratio.

### 3.2.3. Thermal Resistance

The thermal resistance of the counter flow net channel configurations against various concentration ratios and a constant flow rate of 0.001 kg/s is shown in Figure 17. According to the data, the lowest thermal resistance was observed by using a counter flow net channel configuration with a flow rate of 90% in the upper layer and 10% in the lower layer to be $0.207 \, \mathrm{°C/W^{-1}}$, whereas the maximum thermal resistance was observed by the conventional mini channel and reached up to $0.427 \, \mathrm{°C/W^{-1}}$ at the same mass low rate. As a result, higher flow rates should be used for the upper layer to enable more heat movement over it since it has lower thermal resistance than lower mass flow rates. It is also worth noting that since the heat capacity of water is high, the moderated reduction in thermal resistance with increasing the channel layers did not yield significant increase in the outlet temperatures.

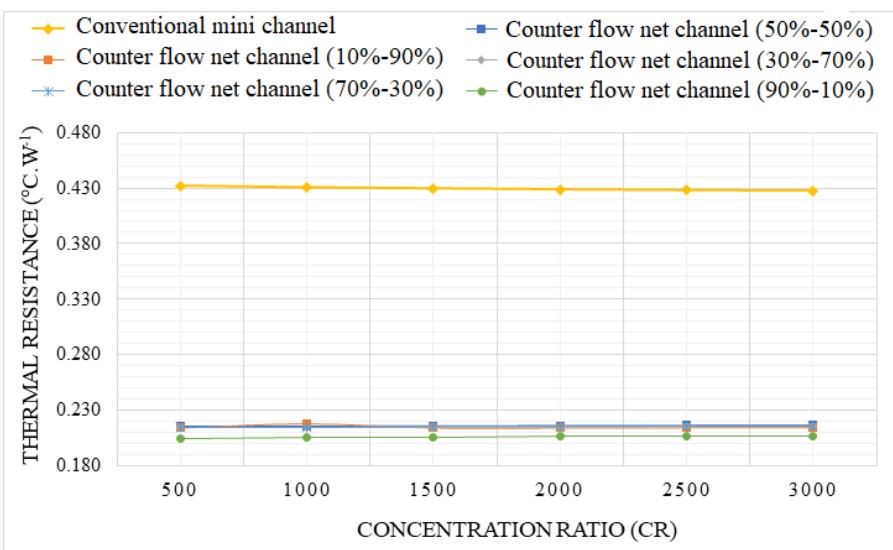

**Figure 17.** Variation of the thermal resistance with different concentration ratio.

### 3.3. Comparison between the Proposed Designs under the Climate Conditions of Dammam City

The performance of the two proposed designs besides the conventional mini channel based-heat sink are analyzed and discussed through this section. Only the configurations of the parallel flow net channel with five layers and the counter-flow double-layered net channel with a flow rate of 90% in the upper layer and 10% in the lower layer are considered in this comparative analysis. For this comparison, the average direct normal irradiance, wind pattern (velocity and rose) and ambient temperature of Dammam city (Given in Table 3) are used and applied to the developed numerical model. The hourly data basis is recorded by the meteorological station implemented at Imam Abdulrahman Bin Faisal University in Dammam city. The average values were obtained by averaging the recorded data from sunrise to sunset during the entire period over the years 2015–2018.

**Table 3.** Average and maximum metrological data of Dammam city.

| Metrological Data | Value |
|---|---|
| Average direct normal irradiance | $428\,\mathrm{W/m^{-2}}$ |
| Maximum direct normal irradiance | $1011\,\mathrm{W/m^{-2}}$ |
| Average wind speed | $3.46\,\mathrm{m/s^{-1}}$ |
| Average wind direction | $8.7°$ |
| Minimum wind speed | $0.3\,\mathrm{m/s^{-1}}$ |
| Average ambient temperature | $30\,°\mathrm{C}$ |
| Maximum ambient temperature | $46.1\,°\mathrm{C}$ |

#### 3.3.1. Maximum Cell Surface Temperature

The variation in maximum cell surface temperature for the three channel heat sink designs is shown in Figure 18, at concentration ratios ranging from $500\times$ to $3000\times$. In these simulations, the convective heat transfer coefficient of the air is dependent on the characteristic lengths, sketch, and roughness of the external boundaries of the geometry design and also on the hydrodynamic characteristics of the wind flow. The figure shows that at the maximum concentration ratio of $3000\times$, the maximum cell surface temperature is minimum for the counter flow net channel heat sink, with a reduction of 11.72% and 12.01% compared to the parallel flow net channel and conventional mini channel, respectively. For the counter flow net channel, the maximum cell temperature does not exceed $55.04\,°\mathrm{C}$ at a concentration ratio of $3000\times$.

### 3.3.2. Outlet Water Temperature

The variation in outlet water temperature is shown in Figure 19. The outlet water temperature is lowest for the counter flow net channel among the selected heat sink designs. This is because in a counter flow configuration, hot water exiting via the lower layer of the heat sink transfers heat to cold water entering through the top layer. When compared to the counter-flow net channel, the conventional and parallel flow net channels have 27.36% and 27.55% higher outlet temperatures, respectively, which makes the conventional channel or net channel superior if possible uses such as desalination are prioritized with the thermal energy recovered from the HCPV system.

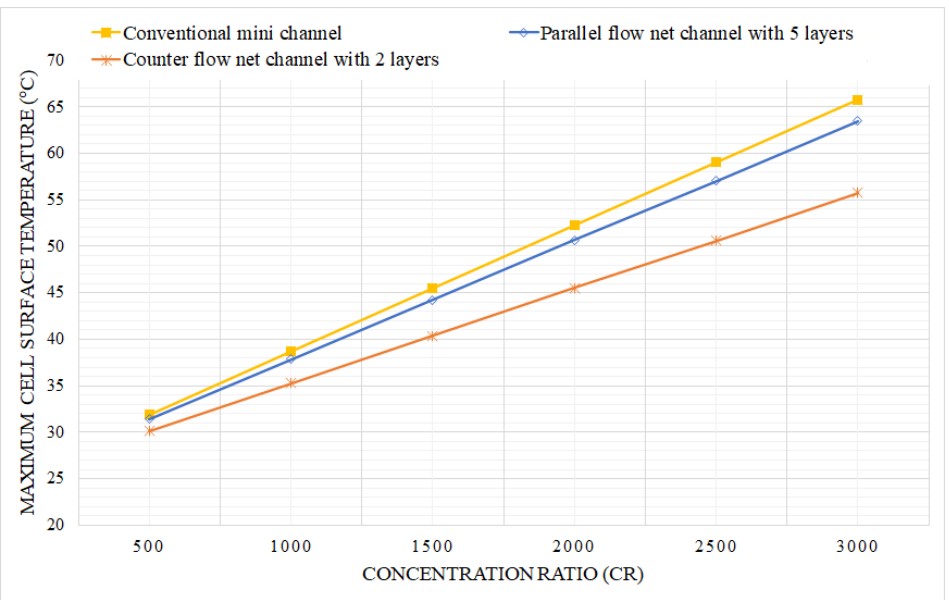

**Figure 18.** Variation of the maximum temperature of the CPV solar cell with different concentration ratio for the conventional channel, net channel with 5 layers and counter flow net channel with a flow rate of 90% in the upper layer and 10% in the lower layer.

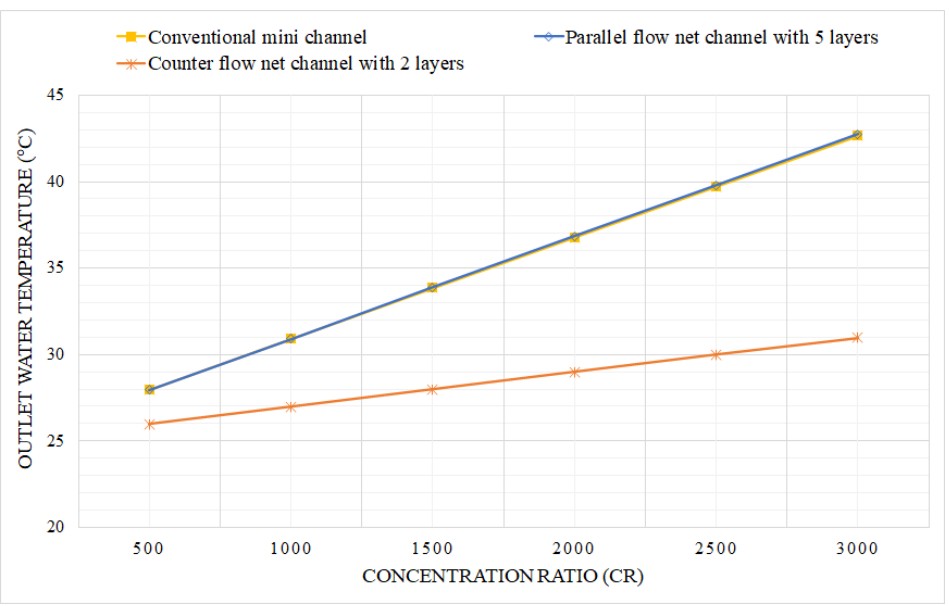

**Figure 19.** Variation of the water outlet temperature of the CPV solar cell with different concentration ratio for the conventional channel, net channel with 5 layers and counter flow net channel with a flow rate of 90% in the upper layer and 10% in the lower layer.

### 3.3.3. Electrical Performance

To analyze the electrical performance of the HCPV module, the electrical efficiency is calculated for the module integrated with all the three heat sink designs and plotted for different concentration ratios as shown in Figure 20. At a water mass flow rate of $0.001 \, kg/s^{-1}$, the HCPV module with the counter-flow double-layered net channel heat sink offers the highest electrical efficiency at all concentration ratios. The difference is considerable at higher concentration ratios, with an increase of 4.42% and 5.83% compared to the parallel flow net channel and conventional channel designs, respectively, at $3000\times$.

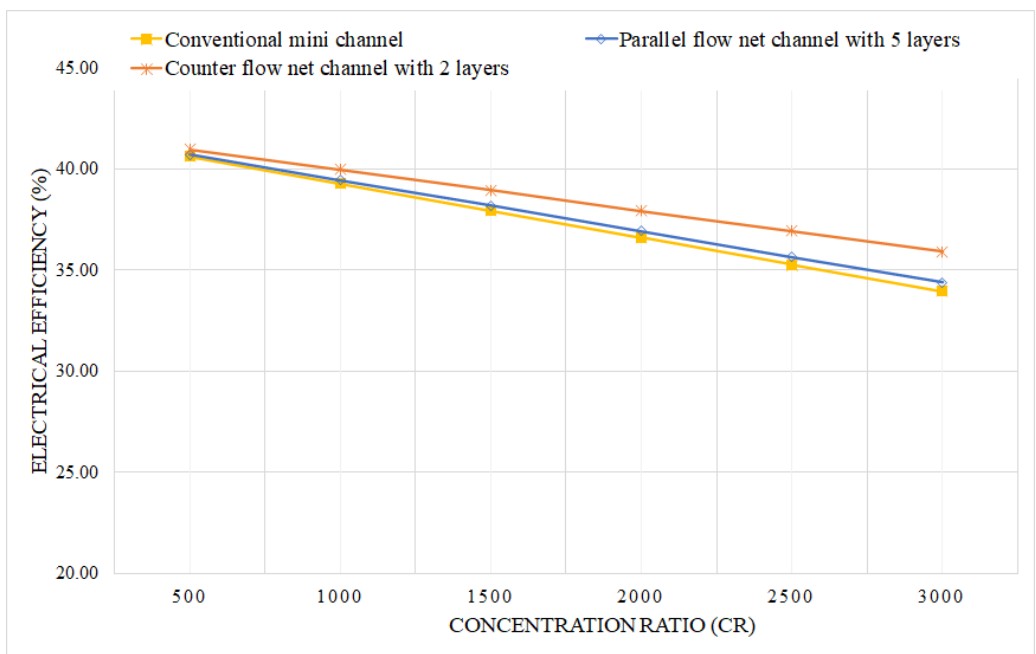

**Figure 20.** Variation of the electrical efficiency of the CPV solar cell with different concentration ratio for the conventional channel, net channel with 5 layers and counter flow net channel with a flow rate of 90% in the upper layer and 10% in the lower layer.

### 3.3.4. Average Thermal Resistance

In this section, the average thermal resistance has been selected to asses the cooling performance of the different net channels and mini channel-based heat sink devices. The average thermal resistance for each mini channel based-heat sink design is shown in Table 4. It can be noted that the lowest thermal resistance is obtained by the counter flow net channel based-heat sink design with an average thermal resistance of $0.22 \, °C/W^{-1}$. This is equivalent to a decrease of 9.30% and 48.83% compared to the parallel flow net channel and counter flow net channel, respectively.

**Table 4.** Average values of thermal resistance for the three heat sink designs for concentration ratios from $500\times$ to $3000\times$.

| Heat Sink Design | Average Thermal Resistance ($°C/W^{-1}$) |
| --- | --- |
| Conventional Channel | 0.43 |
| Parallel flow net channel with 5 layers | 0.39 |
| Counter flow net channel with 2 layers | 0.22 |

## 4. Conclusions

In the present study, new net channel based-heat sink deigns have been investigated. The influence of the number of layers, mass flow rate fraction in each layer, and the coolant flow direction have been tested and discussed. A conventional mini channel based-heat sink was taken as the baseline design for this study. The electrical and thermal

resistance of the parallel-flow and counter-flow net channels for cooling HCPV modules are explored. To identify the optimal configuration of each proposed heat sink design, several configurations of the parallel-flow and double-layered counter-flow net channel are studied. The findings and shortcomings demonstrated that among the tested configurations for the parallel-flow net channels, the net channel with five layers performed better than the other configurations since it yielded the highest outlet water temperature, reaching up 66.49 °C at a concentration ratio of 3000×. This range of temperature can be effectively dedicated for membrane desalination usability, which makes from the net channel with five layers the most appropriate design for such application. Moreover, it has shown its upper hands on ensuring the most uniform and lowest cell temperature relatively compared to the other parallel flow net channels and conventional mini channel. A counter flow double-layered net channel heat sink with five distinct configurations has been selected to study the effect of varied percentages of coolant mass flow rates through the layers of the channel. The counter flow net channel with equal flow rates has achieved a significant reduction in maximum cell surface temperature ranging from 15.43 °C to 22.81 °C when compared to the conventional channel at concentration ratios of 2000× and 3000×, respectively. The lowest thermal resistance was achieved by using a counter flow net channel configuration with a mass flow rate fraction of 90% in the upper layer and 10% in the lower layer to be $0.207\,°C/W^{-1}$.

To map the potential use and check the superiority of the parallel-flow with five layers and the counter flow with a flow rate of 90% in the upper layer and 10% in the lower layer over conventional mini channel based-heat sink designs in Dammam city, a performance chart of the optimum configurations has been discussed in the last section from this paper. On one hand, the counter flow net channel is the optimum design able to maintain the solar cell operating at minimum of temperature ranges. In comparison with the parallel flow net channel with five layers and conventional mini channel, a decrease of 11.72% and 12.01% in maximum cell temperature has been recorded, respectively. Moreover, the use of the counter-flow double-layered net channel heat sink has yielded an electrical efficiency increase of 4.42% and 5.83% compared to the parallel flow net channel and conventional channel designs, respectively, at 3000×. Also, the average thermal resistance of the counter flow net channel is less than the parallel flow net channel and mini channel by 9.30% and 48.83% at 3000×. In comparison to the conventional and parallel flow net channel designs, the counter flow net channel produced the lowest output water temperature with a reduction of 27.36% and 27.55%. This outcome nominates the parallel flow net channel with five layers to be effectively combined with membrane desalination units.

**Author Contributions:** Conceptualization, F.G.A.-A. and T.M.; methodology, F.G.A.-A. and T.M.; software, A.T.O., T.M. and A.K.A.; validation, A.T.O., A.K.A. and F.G.A.-A.; formal analysis, T.M. and R.Z.; investigation, A.T.O., A.K.A. and F.G.A.-A.; resources, the meteorological station implemented at Imam Abdulrahman Bin Faisal University in Dammam city; data curation, the meteorological station implemented at Imam Abdulrahman Bin Faisal University in Dammam city; writing—original draft preparation, A.T.O. and A.K.A.; writing—review F.G.A.-A. and T.M. editing, T.M. and R.Z.; visualization, F.G.A.-A., T.M. and R.Z.; supervision, F.G.A.-A. and T.M.; project administration, F.G.A.-A. and T.M.; funding acquisition, project number IF-2020-024-Eng. All authors have read and agreed to the published version of the manuscript.

**Funding:** This research was funded by the Deputyship for Research & Innovation, Ministry of Education in Saudi Arabia grant project number IF-2020-024-Eng.

**Institutional Review Board Statement:** Not applicable.

**Informed Consent Statement:** Not applicable.

**Data Availability Statement:** The data that support the findings of this study are available on request from the corresponding author. The data are not publicly available due to privacy or ethical restrictions.

**Acknowledgments:** The authors extend their appreciation to the Deputyship for Research & Innovation, Ministry of Education in Saudi Arabia for funding this research work through the project number IF-2020-024-Eng at Imam Abdulrahman bin Faisal University/College of Engineering.

**Conflicts of Interest:** The authors declare no conflict of interest.

## Nomenclature

**Abbreviations**

| | |
|---|---|
| CR | Concentration ratio |
| DNI | Direct normal irradiance |
| GCR | Geometric concentration ratio |
| HCPV | High concentration photovoltaic |

**Symbols**

| | |
|---|---|
| $\dot{V}$ | Volume flow rate [m$^3$/s$^{-1}$] |
| $A_{\text{cell}}$ | Area of the solar cell [m$^2$] |
| $C_p$ | Specific heat capacity [W/m$^{-2}$/°C$^{-1}$] |
| $E$ | Electric power [W] |
| $F_{\text{V}}$ | Body force vector [N] |
| $P$ | Pressure [Pa] |
| $Q$ | Thermal power [W] |
| $q_{\text{con}}$ | Thermal flux via conduction [W/m$^{-2}$] |
| $q_{\text{r}}$ | Thermal flux via radiation [W/m$^{-2}$] |
| $T$ | Temperature [°C] |
| $u$ | Velocity vector [m/s$^{-1}$] |

**Greek symbols**

| | |
|---|---|
| $\alpha_{\text{p}}$ | Thermal expansion coefficient |
| $\eta_{\text{opt}}$ | Optical efficiency of the solar cell |
| $\eta_{\text{sc}}$ | Electrical efficiency of the solar cell |
| $\rho$ | Density [kg/m$^{-3}$] |
| $\tau$ | Viscous stress tensor |

**Subscripts**

| | |
|---|---|
| f, out | Fluid outlet |
| f, in | Fluid inlet |

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
