# Peer review of "Enhanced Net Channel Based-Heat Sink Designs for Cooling of High Concentration Photovoltaic (HCPV) Systems in Dammam City"

_sustainability, doi:10.3390/su14074142_

Round 1

Reviewer 1 Report

General comments:

In this study, authors propose and evaluate active cooling strategies for high concentration PV systems. Different heat sink configurations are investigated in with a specific georeferencing in Saudi Arabia. They studied configurations of net channel based-heat sink designs (channel number, parallel flow, counter flow, mass flow rate, etc.). They conclude with comments on the compromise between efficiency and heat recovery of the overall system.

The paper is nicely written; literature review, assumptions, inputs are clearly explained. The numerical study is sound enough. 

Detailed comments:

Line 187 and line 239: please briefly mention the computational resources used (software version, average CPU time/computational time, hardware configuration) for the repeatability of study.

Line 224: could you please justify the choice of h=15 W/mK ?

Line 250: Good validations. Though, to my opinion, 3M cells is quite high for such configuration (flow is laminar). Did you check near-wall mesh and treatment at lower cell numbers ?

Line 301: not clear, difficult to see the difference between different net channels. Is this mean temperature ?

Line 338: Same comment, difficult to see the difference between 50%-50% and 70%-30%.

Line 462: replace ‘lesser’ by ‘less’.

Line 463: please check the sentence, it seems incomplete.

Reviewer 2 Report

Dear Editor,

Thanks for the opportunity to review the manuscript titled " Novel net channel based-heat sink designs for cooling of high concentration photovoltaic (HCPV) systems in Dammam city”. Solar photovoltaic as clean and green energy technology plays a vital role to fulfil the power shortage of any country. One of the most important obstructions that affect negatively on the electrical efficiency of the solar cell is the magnificent increasing in the temperature of the photovoltaic cell due to the high concentration of solar radiation. So, recent researches have been focused on using heat sink for cooling of high concentration photovoltaic cell. The current paper has suggested a system of net channel based-heat sink designs that try to overcome the increasing in the temperature of the photovoltaic cell. After the reviewing the whole manuscript, there are some notes which need to be considered before publishing this manuscript as in following 

  • The authors addressed that they suggested a " Novel net channel based-heat sink designs", but actually the multi-minichaneel heat sink with both parallel flow and counter flow, one layer or multi-layers have been studied previously in many researches. So, I suggest omitting the "Novel " word from the manuscript.

  • The authors did not mention the type of mesh that used in the suggested model. Besides, it is needed to add a figure for the meshed geometry.

  • The authors stated that the solution of governing equations is applied within a steady state. However, as it is known that the solar irradiance intensity changes with the daytime. The authors should explain how they deal with this case. Besides, the sun irradiance magnitudes of Dammam city need to be included.

  • The authors showed that when increasing the channel layers, the heat sink performed better, this is axiomatical result, because the strategy that used by authors to enhance the thermal performance of minichannel is concentrated on reducing the hydraulic diameter when increasing the layers of channels. For example,

The hydraulic diameter for one layer channel is 0.909 mm, while for fife layers is 0.545mm . That is mean that Nusselt number will increase as the hydraulic diameter decreased. So, if the authors want to know the effect of adding the number of layers , the have to add a channel layers with same hydraulic diameter .

  • The authors neglected the most important parameter that associated with decreasing the hydraulic diameter of minichannel which is the high pressure drop. According to the previous literatures, both the thermal performance and the pressure drop of mini-channel heat sink are inversely proportional with hydraulic diameter. That is mean, the decreasing in hydraulic diameter led to higher pumping power consumption. So, I suggest including the pumping power (or pressure drop) in the comparison between the performance of multi-layers minichannel heat sink.
  • The authors have studied the surface temperature distribution of the solar cell, but I didn’t see any temperature contours that showed the differences in temperature distribution among different designs. So, I suggest including the temperature contours in result section to argue their outcomes.
  • In figure 10, (the Variation of the outlet water temperature of the CPV solar cell with different concentration ratio), it can be seen that a very small differences between the outlet water temperature of various net layers. At the same time, in figure 11, (Variation of the thermal resistance with different concentration ratio) , shows a distinct differences between the thermal resistance of various net layers. that is mean the same of amount of heat extraction from fig.10, but fig.11 shows a significant reduction t in thermal resistance with increasing the channel layers. Can the authors explain the reason?
  • In section 3-3 (Comparison between the proposed designs under the climate conditions of Dammam city) the authors compare between the counter-flow double-layered net channel with equal flow rates and the parallel flow net channel with 5 layers. So, I am asking why the authors didn’t choose the counter flow net channel configuration with a flow rate of 90 % in the upper layer and 10 % which showed the best thermal performance to compare with.

Round 2

Reviewer 2 Report

Dear editor
I am satisfied with the most of author’s responses to my comments raised in my initial review except one comments that must be considered as in the following :
The authors should add the temperature contours of the CPV solar cell for different concentration ratios in order to support their outcomes.

Author Response

Title: Novel net channel based-heat sink designs for cooling of high concentration photovoltaic (HCPV) systems in Dammam city

Ref. No.: sustainability-1637621

Response to the reviewers’ comments as received on 21 March 2022

We sincerely thank the editor and the reviewers for their valuable comments and inputs. The manuscript has been very carefully revised based on the comments. In the revised manuscript, the corrections made in the text are shown in red. The comments and our responses are given below. The line numbers correspond to the ‘marked’ version of the revised manuscript. We hope that the editor and the reviewers will find the revised manuscript suitable for publication.

Reviewer comment:

I am satisfied with the most of author’s responses to my comments raised in my initial review except one comments that must be considered as in the following :

The authors should add the temperature contours of the CPV solar cell for different concentration ratios in order to support their outcomes.

Reply:

The temperature contours of the CPV solar cell for different concentration ratios are added as requested (Please see lines 309-322..
